# Advancements in Glioma Care: Focus on Emerging Neurosurgical Techniques

**DOI:** 10.3390/biomedicines12010008

**Published:** 2023-12-20

**Authors:** Matteo De Simone, Valeria Conti, Giuseppina Palermo, Lucio De Maria, Giorgio Iaconetta

**Affiliations:** 1Department of Medicine, Surgery and Dentistry “Scuola Medica Salernitana”, University of Salerno, Via S. Allende, 84081 Baronissi, Italy; vconti@unisa.it (V.C.); g.palermo10@studenti.unisa.it (G.P.); giaconetta@unisa.it (G.I.); 2Clinical Pharmacology and Pharmacogenetics Unit, University Hospital “San Giovanni di Dio e Ruggi, D’Aragona”, 84131 Salerno, Italy; 3Unit of Neurosurgery, Department of Surgical Specialties, Radiological Sciences, and Public Health, University of Brescia, 25123 Brescia, Italy; lucio.demaria@hcuge.ch; 4Unit of Neurosurgery, Department of Clinical Neuroscience, Geneva University Hospitals (HUG), 1205 Geneva, Switzerland; 5Neurosurgery Unit, University Hospital “San Giovanni di Dio e Ruggi, D’Aragona”, 84131 Salerno, Italy

**Keywords:** gliomas, LITT, brain tumors, FUS, classification, functional neurosurgery

## Abstract

Background: Despite significant advances in understanding the molecular pathways of glioma, translating this knowledge into effective long-term solutions remains a challenge. Indeed, gliomas pose a significant challenge to neurosurgical oncology because of their diverse histopathological features, genetic heterogeneity, and clinical manifestations. Relevant sections: This study focuses on glioma complexity by reviewing recent advances in their management, also considering new classification systems and emerging neurosurgical techniques. To bridge the gap between new neurosurgical approaches and standards of care, the importance of molecular diagnosis and the use of techniques such as laser interstitial thermal therapy (LITT) and focused ultrasound (FUS) are emphasized, exploring how the integration of molecular knowledge with emerging neurosurgical approaches can personalize and improve the treatment of gliomas. Conclusions: The choice between LITT and FUS should be tailored to each case, considering factors such as tumor characteristics and patient health. LITT is favored for larger, complex tumors, while FUS is standard for smaller, deep-seated ones. Both techniques are equally effective for small and superficial tumors. Our study provides clear guidance for treating pediatric low-grade gliomas and highlights the crucial roles of LITT and FUS in managing high-grade gliomas in adults. This research sets the stage for improved patient care and future developments in the field of neurosurgery.

## 1. Introduction

Gliomas, primary brain tumors that originate from glial cells, represent a significant challenge in the field of neuro-oncology. These intra-axial tumors exhibit high variability in their histopathological features, genetic profiles, and clinical manifestations, aspects that make it crucial to take an integrated, multidisciplinary approach to effectively combat their annihilating effect on patients [1].

Over the years, extensive research has shed light on the molecular basis of gliomas, offering promising possibilities for targeted therapies. In addition, advances in neurosurgical techniques have opened new horizons in the management of these tumors, providing more precise and personalized therapeutic options [1,2].

Despite these significant advances in understanding the complexity of glioma molecular pathways, the current standard of care, which includes maximal safe resection followed by radiotherapy and chemotherapy, often fails to provide patients with long-term survival and optimal quality of life. New therapeutic strategies are still needed to address the complexity of glioma biology and improve patient outcomes [2].

This study aimed to comprehensively review the most recent advances in the management of gliomas, focusing on the new classification system and emerging functional neurosurgical techniques. By integrating cutting-edge molecular knowledge with emerging and innovative neurosurgical approaches, it will be possible to lay the foundation for a more holistic and tailored glioma treatment paradigm [2].

## 2. Epidemiology and Classification of Gliomas

Gliomas account for about 80% of all malignant brain tumors, with an incidence that increases with age and peaks in individuals older than 65 years. In the United States, the age-adjusted glioma incidence rate is 6.16 per 100,000 person-years in subjects aged 65 and older, compared with 0.50 per 100,000 person-years in subjects aged 20–44 years [3,4]. Another relevant factor is the origin of the patients. In Europe, the highest incidence rates have been reported in Denmark and Finland, with age-standardized rates of 6.8 and 5.5 per 100,000 person-years, respectively [5]. In the United States, the incidence of gliomas is higher among whites compared with other racial/ethnic groups [6]. Genetic predispositions, such as specific markers associated with glioma susceptibility, contribute to these disparities [7]. Environmental factors, including exposure to ionizing radiation, are implicated in the etiology of GBM [8].

Differences in health infrastructure and diagnostic accessibility also influence reported incidence rates. Developed countries with advanced diagnostic capabilities can detect and report cases more accurately, potentially contributing to higher incidence rates. Conversely, underdiagnosis in some regions may lead to biased reporting [9].

Lifestyle and diet-related factors may further contribute to the complex epidemiological landscape of GBM. Emerging research suggests potential links between certain dietary components and glioblastoma risk. [10]

In essence, the global distribution of glioblastoma involves a complex interplay of genetic, environmental, and health factors. Unraveling these complexities is essential to advancing our knowledge of glioma epidemiology and ultimately improving prevention and treatment strategies on a global scale.

Gliomas can be classified into grades based on their histological features and molecular characteristics. Grade I gliomas are considered benign tumors, while grades II, III, and IV are malignant. The most common malignant gliomas, approximately 50% of all, are grade IV glioblastomas (GBMs), which are most frequently diagnosed in individuals older than 65 years. In the United States, the age-adjusted incidence rate of GBM is 3.21 per 100,000 person-years in subjects aged 65 and older, compared with 0.23 per 100,000 person-years in individuals aged 20–44 years [3].

Recent advances in molecular profiling have led to a better understanding of the underlying genetic alterations that drive the development of gliomas and can be used to classify them into molecular subtypes, which are characterized by different clinical outcomes and responses to treatment.

Among the most important genetic variants, mutations in the isocitrate dehydrogenase (IDH) gene identify IDH-mutant gliomas that demonstrate a better prognosis than wildtype IDH gliomas [11,12]. Other molecular alterations include mutations in the tumor-suppressor gene tumor protein 53 (TP53), the alpha-thalassemia/mental retardation x-linked (ATRX) syndrome, and the epidermal growth factor receptor (EGFR) pathway [13]. However, even taking into account this valuable genetic profiling, establishing glioma prognosis is more complicated because it is associated with a complex network of factors that contribute to determining the likelihood of patient survival [14].

### 2.1. Classification of Gliomas

In 2021, the World Health Organization (WHO) provided a new classification of tumors of the central nervous system, which is based on a more comprehensive understanding of the molecular and genetic characteristics of tumors and suggests the use of a combination of histology and molecular markers to predict patient outcomes and guide the choice of appropriate treatments (Table 1).

In particular, the document identifies four categories of gliomas: (1) diffuse astrocytic and oligodendroglial tumors, which include diffuse astrocytoma IDH-mutant, anaplastic astrocytoma IDH-mutant, oligodendroglioma IDH-mutant, and 1p/19q co-deleted tumors; (2) anaplastic astrocytic and oligodendroglial tumors, including anaplastic astrocytoma IDH-mutant, anaplastic oligodendroglioma IDH-mutant, and 1p/19q co-deleted tumors; (3) GBMs, including GBM IDH-wildtype and GBM IDH-mutant; (4) and other gliomas, such as ependymoma, choroid plexus tumors, and embryonal tumors.

The WHO classification does not automatically associate histological type with the grade of malignancy. For example, in the past, anaplastic astrocytoma was automatically considered a grade III glioma, as was anaplastic meningioma.

Therefore, it was expected that these two tumors, although biologically different, would have similar survival times. However, this is a simplistic approach, and today, the method is to stratify the various histological types as much as possible and to study their characteristics in more detail [15].

### 2.2. Molecular Signature of Gliomas

Accurately classifying the different types of gliomas by combining specific genetic and molecular features is critical to mitigating the difficulties of treating such a heterogeneous group of malignancies [16,17] and maximizing the chances of success [18].

Indeed, the molecular signature is strongly associated with the pathogenesis and prognosis of several tumors. Glioma pathogenesis is no exception, as it is closely dependent on genetic and epigenetic alterations, cellular signaling pathways, and the tumor microenvironment. The PI3K-Akt-mTOR pathway, which modulates cell growth, proliferation, survival, and metabolism, is one of the most important signaling pathways in glioma pathogenesis. As suggested by Wang et al., this pathway leads to increased activity in downstream effectors that promote glioma growth and progression through the amplification of growth factor receptors or the loss of negative regulators, promoting the processes of invasion and metastasis and resistance to chemotherapy and radiotherapy [19]. Of note, according to Cancer Genome Atlas (TCGA) data, approximately 88% of diffuse gliomas, which include GBM and lower-grade gliomas (LGGs), have genetic alterations in at least one component of the PI3K-Akt-mTOR pathway [15]. Specifically, mutations in the gene encoding the catalytic subunit of PI3K (PIK3CA) and the regulatory subunit of PI3K (PIK3R1) were found with a frequency of 17% and 18%, respectively. Less frequent are mutations in the downstream effector Akt (AKT1) and the gene encoding the component of the mTOR complex (MTOR), identified in 2% and 3% of diffuse gliomas, respectively. In addition, amplification of the gene encoding the epithelial growth factor receptor (EGFR), a potent activator of the PI3K-Akt-mTOR pathway, observed in approximately 50% of GBMs, further underscores the importance of this pathway in glioma pathogenesis [20,21,22]. As a result, targeting this signaling pathway is now considered a promising therapeutic strategy for treating gliomas, with several drugs currently in clinical trials [23,24]. Table 2 summarizes the most studied mutations in gliomas with their relative frequency and etiopathogenetic roles. The frequency of each genetic variant is represented by a wide range because of the different grades and histological types of gliomas considered.

Lei et al. developed a model using transcriptome data from two cohorts of patients with GBM. They identified 341 metabolic genes that showed significant differences between normal brain and GBM tissues, among which, 56 genes were found to be correlated with the patients’ overall survival (OS). In the end, the model was constructed using a Lasso regression model with 18 genes and showed high accuracy in predicting the OS of the patients. In particular, the high-risk group of patients included in this model had a significantly shorter OS than the low-risk group in the training cohort (*p* < 0.0001) and in the independent external validation (*p* < 0.001). The study by Lei et al. showed that the prognosis of GBM is closely related to metabolic pathways and that, by using a model, it is possible to predict the prognosis of patients with GBM [41]. As is evident in contemporary neuro-oncology, searching for better diagnostic and therapeutic strategies for glioblastoma has led to a deep exploration of prognostic and predictive biomarkers. Increasingly, the evidence allows us to make a targeted choice regarding the type of treatment in relation to the molecular profile of the tumor.

In fact, these biomarkers may now influence the choice of therapeutic approach. For example, the presence of the mutated IDH1 gene is associated with a better prognosis, and patients carrying this mutation may benefit from more aggressive surgical resection aimed at maximal tumor excision [42].

MGMT promoter methylation status is another crucial biomarker. Glioblastoma patients with MGMT promoter methylation tend to respond better to chemotherapy [43], and knowledge of this status may influence the decision on the extent of surgical resection and the subsequent use of adjuvant therapies. The expression of certain molecular markers, such as EGFR, may also impact the choice of treatment. Elevated EGFR expression may suggest a more aggressive tumor phenotype, influencing the decision for a more extensive surgical approach or the inclusion of targeted therapies [44]. Montano et al. observed significantly longer OS in GBM patients with high levels of EGFR treated with total resection and standard radiochemotherapy involving temozolomide [45].

An innovative approach to glioblastoma can be achieved not only through research on new biomarkers but also through new treatment technologies.

Among the innovative avenues gaining traction is the study of PROteolysis TAgeting Chimeras (PROTACs) as potential game-changers in glioblastoma therapy [46].

PROTACs represent a paradigm shift in drug development, harnessing the cellular machinery to induce targeted protein degradation. In the context of glioblastoma, this approach holds the promise of selectively eliminating specific oncogenic proteins, thereby disrupting key pathways implicated in tumor progression. Yang et al. used a therapeutic nanosystem created by combining the BRD4-degrading PROTAC ARV-825 with a complex micelle. This micelle was able to penetrate the blood–brain barrier and target brain tumors. The drug released by this system shows antitumor effects by reducing cell proliferation, inducing apoptosis, and suppressing M2 macrophage polarization. These effects are achieved through the inhibition of IRF4 promoter transcription and the phosphorylation of STAT6, STAT3, and AKT [47].

To contextualize these advancements, a comprehensive understanding of the molecular landscape of glioblastoma subtypes is imperative. This molecular intricacy necessitates a tailored approach, recognizing that one-size-fits-all interventions may fall short in addressing the diverse manifestations of the disease.

Neurosurgical approaches and treatment modalities are intrinsically linked to the molecular profile of the glioblastoma, and understanding these connections is pivotal in optimizing patient outcomes. The choice between maximal safe resection, adjuvant therapies, and targeted interventions hinges on a nuanced appreciation of the specific molecular signatures at play [48]. Given the complex interplay between biomarkers, neurosurgical strategies, and other therapeutic options, we approach a future in which personalized and precise interventions will redefine the trajectory of glioblastoma management.

## 3. Overview of Treatments for Gliomas

Surgery has been an important component of the management of gliomas since the late 19th century. The first surgical resection was performed in 1884 by Victor Horsley, who removed a frontal lobe glioma from a patient who had been experiencing seizures. On 25 November 1884, Mr. Rickman J. Godlee performed the first recognized resection of a primary brain tumor. The surgery was performed at the Hospital for Epilepsy and Paralysis in London, UK. The patient died postoperatively from apparent meningitis, but postmortem examination revealed no residuals of the excised glioma [49].

In the following decades, surgical techniques have continued to evolve, with the development of new tools and methods to access and remove brain tumors.

In the early 20th century, Harvey Cushing became one of the most prominent neurosurgeons contributing significantly to glioma surgery. He is credited with developing the transsphenoidal approach, a minimally invasive method of accessing tumors located at the base of the skull. He also introduced the use of the operating microscope, which enabled more precise and controlled surgical resections [50].

In the mid-20th century, the use of neuroimaging techniques such as computed tomography (CT) and magnetic resonance imaging (MRI) revolutionized the diagnosis and management of gliomas. These techniques enabled more accurate preoperative planning and intraoperative visualization of the tumor and surrounding brain tissue [51].

Many advances in glioma surgery have been made in recent decades, including the development of intraoperative neurophysiological monitoring, which enables the real-time monitoring of neurological function during surgery. This technique has helped to minimize the risk of neurological deficits associated with the surgical resection of gliomas [52].

In addition, image-guided navigation systems and endoscopic techniques have expanded the range of accessible tumors and increased the precision of surgical resection [53,54].

Despite these advances, surgical resection of gliomas remains a complex and challenging procedure, particularly for high-grade tumors located in critical or eloquent areas of the brain. Furthermore, for these cases, the development of new surgical tools, such as ultrasonic aspirators and lasers, has enabled more effective and efficient tumor removal. In recent years, interest has grown in the use of less invasive approaches, such as stereotactic radiosurgery and laser interstitial thermal therapy (LITT). These approaches may offer a less invasive alternative to surgical resection for certain types of tumors [55,56].

The primary goal of surgery is to safely remove as much of the glioma as possible while preserving neurological function. The extent of surgical resection is a major determinant of patient survival and functional outcomes [57].

Several factors should be considered when deciding on the optimal surgical approach to gliomas, including the location, size, and grade of the tumor, as well as the patient’s age and general health status [58]. In general, surgical resection is recommended for patients with low-grade gliomas and for those with high-grade gliomas who can tolerate the procedure [59].

The goal of surgery for high-grade gliomas is typically to achieve a safe maximal resection (SMR), which involves removing as much of the visible tumor as possible while preserving neurological function [60]. The extent of resection is typically assessed using MRI and is classified as gross total resection (GTR), subtotal resection (STR), or partial resection (PR) [61,62].

Several studies have demonstrated that achieving a GTR is associated with improved OS and progression-free survival (PFS) in patients with high-grade gliomas. A meta-analysis of 37 studies found that patients who underwent GTR probably increased the likelihood of 1-year survival compared with STR by about 61% and increased the likelihood of 2-year survival by about 19% (GTR compared with STR at 1 year: RR, 0.62; 95% CI, 0.56–0.69; *p* < 0.001) [63].

However, achieving a GTR is not always possible or safe. Indeed, in the case of tumors located in critical or eloquent areas of the brain case, the goal of surgery may be to obtain a biopsy or to remove the tumor to alleviate symptoms. Increasing evidence has shown that the use of fluorescence-guided surgery, which involves the administration of a fluorescent contrast agent that accumulates in tumor tissue, can improve the extent of glioma resection and increase PFS in patients with high-grade gliomas [64,65]

These studies have demonstrated that patients undergoing fluorescence-guided surgery have a significantly higher rate of complete tumor resection than those undergoing conventional white-light surgery.

Another recent study investigated the impact of intraoperative MRI (iMRI) on the extent of resection and patient outcomes in glioma surgery [60]. The study found that the use of iMRI can improve the extent of tumor resection and increase PFS in patients with high-grade gliomas. In addition, the use of iMRI is associated with a lower risk of postoperative neurological deficits.

Advances in imaging technology have also led to the development of new tools for preoperative planning and intraoperative navigation. For example, diffusion tensor imaging (DTI) has been shown to provide valuable information on the location and orientation of white matter tracts in the brain, which can help surgeons avoid damaging these critical structures during tumor resection [66].

There has also been growing interest in the use of minimally invasive approaches for the treatment of gliomas, such as LITT and stereotactic radiosurgery (SRS). A recent study compared the effectiveness of LITT and SRS for the treatment of recurrent high-grade gliomas and found that both approaches were associated with similar rates of tumor control and survival [67]. The study suggested that LITT may be a less invasive alternative to SRS for certain types of recurrent gliomas.

In summary, recent studies have focused on refining surgical techniques for glioma resection and identifying the most effective surgical strategies to improve patient outcomes. Advances in imaging technology and the development of new tools for operative navigation have also provided valuable insights into improving surgical precision and minimizing damage to surrounding brain tissue.

### The Standard of Care for Gliomas

The standard of care for gliomas depends on various factors, including the grade of the tumor, the location and size of the tumor, and the patient’s health status. However, the current standard of care typically involves a combination of surgery, radiation therapy, and chemotherapy.

Surgery is usually the first-line treatment and aims to remove as much of the tumor as possible while preserving critical brain functions. The extent of tumor resection is an important predictor of patient outcomes, and efforts to maximize resection while minimizing damage to the surrounding brain tissue have led to the development of various surgical techniques and technologies, such as intraoperative imaging and fluorescence-guided surgery [68].

Radiation therapy is typically used following surgery to kill any remaining tumor cells and prevent tumor regrowth. Various types of radiation therapy can be used for gliomas, including external beam radiation therapy, brachytherapy, and stereotactic radiosurgery [69]. The use of concurrent chemotherapy and radiation therapy has also been shown to improve outcomes in certain cases, such as for patients with high-grade gliomas [70].

Chemotherapy is usually reserved for cases where surgery and radiotherapy alone are not sufficient or in cases where the tumor recurs after initial treatment. Several chemotherapy agents can be used for gliomas, such as the DNA alkylating agents temozolomide (TMZ) and carmustine (BCNU) [71] TMZ is the most used drug in the treatment of high-risk gliomas after surgical resection. However, chemoresistance occurs in many patients, representing a substantial obstacle to successful therapy. A crucial role in chemoresistance is played by genes encoding for DNA repair proteins, such as DNA mismatch repair (MMR)-related proteins, O-6-methylguanine-DNA methyltransferase (MGMT), base excision repair (BER)-related proteins, AlkB homolog 2, alpha-ketoglutarate-dependent dioxygenase (ALKBH2), and proteins involved in homologous recombinational repair/non-homologous end joining (HRR/NHEJ), all of which correlate with TMZ efficacy [72].

Fortunately, today, it is possible to use a pharmacogenetically guided approach to predict the degree of sensitivity or resistance to a specific drug [73]. Although there are still barriers to overcome, pharmacogenetics, which studies responses to drug therapy based on a patient’s genetic background, is very useful in personalizing drug therapy in all medical areas [73,74]. Regarding TMZ, pharmacogenetic testing based on the determination of MGMT methylation status is a useful tool for predicting responses to TMZ [71].

In 2016, Buckner et al. analyzed the clinical outcomes of 254 patients younger than 40 years randomized to receive radiotherapy (n. 128) or radiotherapy with chemotherapy (n. 126). The patients had astrocytoma, oligodendroglioma, or oligoastrocytoma and, after STR, were randomly separated into two treatment groups.

The results showed that, regardless of histologic type, patients who received radiotherapy with chemotherapy had a longer median OS than those who received chemotherapy alone (13.3 years vs. 7.8 years; *p* = 0.003). The two OS curves did not separate immediately but rather after one year in astrocytoma and about three years in oligoastrocytoma and oligodendroglioma.

The two curves, in all cases, remained well separated and distinct until the end of observation, about 12 years. The study showed that disease-free survival was also higher in patients who received chemotherapy and radiotherapy than those who received only radiotherapy, but the data are less reliable because these patients were promptly treated in various ways to improve prognosis [75].

The standard of care for gliomas is continuously evolving as new treatment options and strategies are developed. Recent studies have investigated the potential benefits of immunotherapy, targeted therapy, and other therapeutic approaches [76], and clinical trials are ongoing to determine the safety and efficacy of these new treatment options [77].

We report in Table 3 a brief description of each of the treatments for gliomas.

## 4. Functional Neurosurgery

Functional neurosurgery is a rapidly advancing subspecialty that aims to manage neurological disorders through precise surgical interventions targeting specific brain circuits and regions. Unlike conventional neurosurgery, which often focuses on removing lesions or repairing structural abnormalities, functional neurosurgery primarily aims to modulate neural activity or disrupt malfunctioning circuits to restore normal brain function or mitigate symptoms. Two examples of innovative treatments applicable to brain tumors are focused ultrasound surgery (FUS) and LITT [78].

FUS is a non-invasive medical procedure that uses ultrasound waves to heat and destroy targeted tissues within the body without the need for traditional surgery. Focused ultrasound has been researched and developed as a potential treatment option for various medical conditions, including brain tumors. During the procedure, multiple intersecting beams of ultrasound energy are focused precisely on the tumor, generating enough heat to destroy the abnormal tissue while leaving surrounding healthy tissue relatively unharmed [79].

LITT has emerged as a promising treatment option for gliomas, which are difficult to treat with traditional surgery, radiation, and chemotherapy because they often invade surrounding brain tissue. LITT for gliomas is a relatively new approach with promising results. It is often used as an adjunct to traditional treatments, such as surgery and radiotherapy, or as a primary treatment option for smaller tumors. However, not all patients are candidates for LITT, and careful evaluation is necessary to determine the most appropriate treatment plan [80].

### 4.1. Laser Interstitial Thermal Therapy (LITT)

LITT is a minimally invasive surgical technique that uses a laser to create a controlled lesion within the brain and to destroy cancerous or abnormal tissue. It is a type of thermal ablation therapy that delivers controlled heat to the tumor, causing the cancer cells to die.

LITT is performed under general anesthesia to ensure patient comfort and safety during the procedure. The head is placed in a rigid frame, and an MRI is performed. An appropriate trajectory is established, and then, a small incision and straw-sized hole are made in the skull. A thin, flexible laser probe is inserted into the lesion. The laser is then used to heat and destroy the tumor while sparing healthy brain tissue. During the procedure, real-time MRI is used to guide laser placement and monitor treatment progress. After surgery, the patient is closely monitored for neurological deficits or complications. Depending on the case, the patient may require further treatment, such as radiotherapy or chemotherapy [81,82].

The use of lasers for medical applications dates back to the 1960s, but LITT as a therapeutic technique for brain tumors was first proposed in 1983 [83]. However, its use is limited because of an inability to monitor tissue temperature during laser treatment and, thus, control the extent of ablation [84].The first clinical application of LITT for brain tumors was reported in 1992, and since then, the technique has been increasingly used in neurosurgical practice [80]. However, it should be mentioned that LITT is used on many other organs.

The development of LITT has been driven by advancements in laser technology, imaging technology, and computational modeling. The use of MRI or CT to guide laser placement and monitor thermal ablation has enabled the accurate and precise treatment of brain tumors with minimal damage to surrounding tissue. The development of computer simulations of laser–tissue interactions has also improved the safety and efficacy of LITT [81]. In each laser application, absorbed light is converted into heat, causing changes in tissues. Wavelengths between 620 and 1200nm provide low absorption and deep penetration, which are optimal for LITT.

LITT is effective for the treatment of a variety of brain tumors, including GBMs, metastatic tumors, and low-grade gliomas. Studies have demonstrated that LITT is associated with a low complication rate and shorter hospital stays compared with traditional brain surgery. In addition, LITT has been used in the treatment of epilepsy, Parkinson’s disease, and other neurological conditions.

Although LITT has shown promise as a minimally invasive alternative to traditional brain surgery, further research is needed to fully establish its safety and efficacy. Ongoing clinical trials are exploring the use of LITT for the treatment of other neurological conditions and recurrent brain tumors [85].

### 4.2. Indications for LITT

LITT is a relatively new procedure, and guidelines, which vary depending on the institution and the specific case, are still evolving [86,87,88,89,90,91,92,93,94]. Table 4 summarizes some guidelines for LITT.

First, a detailed preoperative evaluation must be performed. On the one hand, it is mandatory to characterize the tumor. LITT is usually considered for patients with small or deep brain tumors, which are not good candidates for traditional surgery or radiotherapy. On the other hand, it is also important to define the patient’s clinical features, paying special attention to their neurological status. This evaluation usually includes a thorough history, physical examination, and imaging studies, such as MRI or CT scans [85].

#### 4.2.1. LITT in Low-Grade Gliomas

Treatments of low-grade gliomas (LGGs) must consider how best to manage symptoms and remove or shrink the tumor. The optimal treatment of LGGs, particularly the timing, is controversial, and decision making must balance the benefits of therapy with potential treatment-related complications.

Although diffuse LGGs account for only 15% of gliomas, they have received increasing attention in the past decade [95]. The standard of care is surgical resection, but it can be difficult to perform radical surgery because of the anatomical location and surrounding functional and vascular structures. Such anatomical complexity often makes nonnegligible surgical sequelae highly likely, so less invasive techniques may confer a more optimal balance between cytoreduction and surgical complications [89,90,91].

In such cases, alternative treatments, including LITT, are preferred. Indeed, even in recurrences, it has already been evident for several years that LITT, given its minimal invasiveness, is the therapy of choice and may be preferred over reoperation [96].

In 2013, Patel et al. set out to analyze changes in lesion volume during the period following LITT using polygonal fusion tracing. Sixteen patients with intracranial neoplasms participated in the study. Using the OsiriX DICOM Viewer, three evaluators calculated lesion volumes as follows: pre-ablation (PreA), immediate post-ablation (IPA), 24 h post-ablation (24PA), and first post-ablation follow-up (FPA), which ranged from 4 to 11 weeks after ablation. The observed sizes showed an acute increase in volume at IPA with a decrease in the size at 24PA. Therefore, it is recommended that conclusions about the size of post-LITT intracranial lesions be drawn at least 24 h after LITT rather than immediately thereafter [97].

In a 2012 study, Jethwa et al. selected 20 patients who had previously failed conventional therapies; were unable to tolerate an open cranial procedure; or, finally, had a tumor considered otherwise inoperable. The patients then underwent LITT. The result was a particularly short hospitalization period of 2.27 days, with most patients going home on the first day postoperative. Complications, on the other hand, occurred in only four patients, generally those who were in poor health before surgery. Thus, the study argues that laser-induced heat therapy is a procedure for difficult intracranial neoplasms. Like any other procedure, it can provide excellent results if patient and lesion selection is performed carefully [98].

This evidence suggests that, even in LGGs, LITT plays a nonnegligible role, so it could not only be an alternative for inoperable cases but also a first choice in the neurosurgical armamentarium, especially in recurrences.

#### 4.2.2. LITT in GBMs

GBM is the most common and aggressive malignant primary brain tumor. Several treatment options are available today, but patients have a median survival of 12–15 months [13]. Surgery is generally considered the mainstay of treatment for patients with surgically accessible tumors; however, GBM has a strong tendency to recur [99].

GBM recurrence is a major problem because subsequent surgical resection is not always feasible and, even if possible, does not always benefit the patient. In other circumstances, patients are generally managed with chemo-radiotherapy, with a worse prognosis. The advent of LITT has provided a potentially viable salvage therapy for patients with recurrent or newly diagnosed lesions inaccessible to conventional surgical approaches. In a study by Traylor et al., 69 patients with a median age of 56 years with GBMs who underwent LITT were retrospectively analyzed. OS from the time of LITT was the primary endpoint measured.

The median tumor volume was 10.4 cm^3^ (range, 1.0–64.0 cm^3^). The Kaplan–Meier estimate from the time of LITT was 12 months (95% confidence interval of 8–16 months). The median volume uncovered by the ablation beam was 1.31 cm^3^ (range, 0–41.2 cm^3^). The median hospital length of stay was two days (range, 0–47 days), and the median PFS from the time of LITT was four months (95% confidence interval, 3–7 months). Adjuvant chemotherapy significantly prolonged PFS and OS (both, *p* < 0.01). Total gross ablation was not significantly associated with PFS (*p* = 0.09).

The results of this study suggest that LITT may confer a survival benefit over the nonoperative management of newly diagnosed GBM. However, the treatment of traditional inoperable GBMs with surgical therapy represents a challenge for physicians, and larger studies are needed to establish the impact of this surgical technique on GBM management [94].

## 5. Focus Ultrasound and LITT: Similarities and Differences

The effectiveness of LITT has already been described, so now, we will report the evidence related to FUS; the possible limitations of the two techniques; and finally, the choice between FUS and LITT with their respective indications.

Although FUS and LITT share some similarities in mechanisms of action, there are also obvious and understandable differences between the two techniques. FUS uses focused ultrasound waves to generate high levels of thermal energy at a precise location in the brain, creating a small lesion that can destroy tumor tissue. The procedure is non-invasive and is performed while the patient is awake, with real-time MRI or ultrasounds to guide the procedure [100]. The clinical indications for FUS are given in Table 5.

Several studies have examined the effectiveness of these surgical approaches in terms of PFS and OS in patients with gliomas. The available literature suggests that LITT and FUS are both safe and effective treatments for such tumors [101].

The literature offers a lot of evidence. Among the most recent, we have selected two highly indicative studies. A study by Coluccia et al. published in 2020 found that patients with GBMs who had not received previous chemotherapy had a longer PFS and OS after treatment with FUS [102].

Furthermore, another study performed in the same years by Deng et al. also reported that patients who had not received previous bevacizumab therapy had a longer PFS and OS after FUS treatment [103].

Since both techniques are effective, the therapeutic choice should be based on the possible difficulties of adopting the techniques. It has emerged from various studies that the effectiveness of LITT is strongly influenced by the size and location of the tumor, making it less effective for large and deep tumors and more effective for small and deep tumors. The location of the tumor means not only a greater or lesser depth within the brain but also proximity to particularly critical anatomical structures or eloquent areas of the brain.

Regarding tumor dimensions, a study by Chang et al. (2021) found that patients with tumors larger than 30 cc had significantly worse PFS and OS after FUS treatment compared with those with smaller tumors [104].

Overall, LITT may be preferable for larger or more diffuse tumors [104], especially those near critical brain structures [105] or in areas responsible for essential functions [98]. FUS is typically chosen for smaller or deep-seated tumors in anatomically less complex regions that cannot be surgically removed [100,106,107,108,109]. FUS and LITT show similar effectiveness for tumors close to the brain’s surface, as well as small ones [100,105].

**Table 5 biomedicines-12-00008-t005:** Indications for FUS.

Patient Age and Glioma Grade	Indications for FUS	References
Pediatric Low-Grade Gliomas	Recurrent or residual tumors that are not amenable to resection or conventional radiation therapy	[106,107]
Adult Low-Grade Gliomas	Small or deep-seated tumors that are not amenable to resection or as an adjunct to surgical resection to improve the extent of resection	[108]
High-Grade Gliomas (Grade III/IV)	As a salvage therapy for recurrent tumors or as a palliative therapy for patients with poor performance status	[109]

In conclusion, both FUS and LITT are generally well tolerated, with relatively low rates of adverse effects such as infections, hemorrhages, or neurological deficits. However, as with any medical procedure, there are potential risks associated with each technique.

## 6. Conclusions and Future Directions

Through advances in molecular diagnosis, clinicians can better understand the underlying genetic and molecular characteristics of gliomas, leading to more personalized treatment strategies. Targeted therapies and precision medicine offer promising avenues for improving the effectiveness of glioma treatment while minimizing side effects.

The emergence of functional neurosurgery techniques, particularly FUS and LITT, represents a significant step forward in the management of gliomas. These minimally invasive procedures offer the potential for precise and controlled tumor ablation, reducing damage to surrounding healthy brain tissue and improving patient recovery.

However, while both have shown promise in achieving local tumor control and improving outcomes for patients with gliomas, further research is needed to determine their long-term effects on OS and to fully understand their risks and benefits. Ultimately, the choice of technique to be used should be made on a case-by-case basis, taking into account several factors such as tumor location, size, and grade, as well as the general health of the patient and treatment goal. Overall, the literature suggests that LITT is preferred for larger, diffuse tumors near critical brain structures or essential functions, while FUS is typically used for smaller, deep-seated tumors in less complex areas. FUS and LITT are equally effective for superficial and small tumors. As a general guideline, both techniques are used in the treatment of pediatric low-grade gliomas for recurrent or inoperable cases.

In adults, these techniques can play a crucial role in the management of high-grade (grade III/IV) gliomas, serving as salvage therapies for recurrent tumors and providing palliative care for patients with reduced performance status.

## Figures and Tables

**Table 1 biomedicines-12-00008-t001:** Comparison of overall survival and progression-free survival of various types of gliomas.

Glioma Classification	Molecular Markers	Histology	Malignancy	OS	PFS	References
Diffuse astrocytic and oligodendroglial tumors	IDH-mutant	Diffuse astrocytoma	Low grade	>5 years	Variable	[15]
		Anaplastic astrocytoma	Intermediate grade	2–5 years	6–12 months	[15]
		Oligodendroglioma with 1p/19q co-deletion	Low grade	>5 years	Variable	[15,16]
Anaplastic astrocytic and oligodendroglial tumors	IDH-mutant	Anaplastic astrocytoma	Intermediate grade	2–5 years	6–12 months	[15,16]
		Anaplastic oligodendroglioma with 1p/19q co-deletion	Intermediate grade	2–5 years	Variable	[15,16]
Glioblastoma	IDH-wildtype	Glioblastoma	High grade	15 months	7–9 months	[15]
	IDH-mutant	Glioblastoma	High grade	31–46 months	11–20 months	[15]

Abbreviations: OS, overall survival; PFS, progression-free survival.

**Table 2 biomedicines-12-00008-t002:** Genes affecting GBM tumor growth with their related mutation frequencies.

Gene	Role in GBM Tumor Growth	Mutation Frequency	References
EGFR	Amplification or mutation of EGFR leads to increased proliferation and invasion of tumor cells.	40–60%	[22,23,24]
PTEN	Loss of PTEN function promotes tumor cell survival and proliferation.	20–40%	[15,25,26]
TP53	Mutation of TP53 is associated with a more aggressive phenotype.	25–30%	[27,28,29]
IDH1	Mutation of IDH1 is associated with a better prognosis.	5–10%	[25,30].
MGMT	Methylation of the MGMT promoter is associated with increased sensitivity to chemotherapy and better patient outcomes.	30–40%	[31,32,33]
VEGF	Overexpression of VEGF promotes angiogenesis and tumor growth in GBM.	50–80%	[34,35,36]
PDGF	Overexpression of PDGF and its receptor promotes tumor cell proliferation and migration.	15–20%	[37,38]
CDK4	Amplification of CDK4 promotes cell cycle progression and tumor growth.	5–10%	[39,40]

Abbreviations: OS, overall survival; PFS, progression-free survival; EGFR, epidermal growth factor receptor; PTEN, phosphatase and tensin homolog; TP53, tumor protein 53; IDH1, isocitrate dehydrogenase 1; MGMT, O6-methylguanine-DNA methyltransferase; VEGF, vascular endothelial growth factor; PDGF, platelet-derived growth factor; CDK4, cyclin-dependent kinase 4.

**Table 3 biomedicines-12-00008-t003:** Treatments for gliomas.

Treatment	Description	Reference
Surgery	First-line treatment to remove as much of the tumor as possible while preserving critical brain functions.	[68]
LITT	A minimally invasive procedure that uses laser energy to heat and destroy tumor cells. It can be used as an alternative to traditional surgery or as a salvage treatment option for recurrent tumors.	[75]
FUS	A non-invasive procedure that uses focused ultrasound waves to heat and destroy tumor cells. It can be used as an alternative to traditional surgery or as a salvage treatment option for recurrent tumors.	[76]
Radiation therapy	Used following surgery to kill any remaining tumor cells and prevent tumor regrowth. Various types of radiation therapy can be used, including external beam radiation therapy, brachytherapy, and stereotactic radiosurgery.	[69]
Chemotherapy	Reserved for cases where surgery and radiation therapy alone are insufficient or for cases where the tumor recurs following initial treatment. Various chemotherapy agents can be used, such as temozolomide and carmustine (BCNU).	[71]
Concurrent chemotherapy and radiation therapy	Used in certain cases, such as for patients with high-grade gliomas, to improve outcomes.	[70]
Immunotherapy	Investigated as a potential treatment option for gliomas.	[77]
Targeted therapy	Investigated as a potential treatment option for gliomas.	[56]

Abbreviations: LITT, laser interstitial thermal therapy; FUS, magnetic resonance-guided focused ultrasound.

**Table 4 biomedicines-12-00008-t004:** Indications for LITT.

Patient Age and Glioma Grade	Indications for LITT	References
Pediatric Low-Grade Gliomas	Recurrent or residual tumors that are not amenable to resection or conventional radiation therapy.	[86,87]
	Newly diagnosed tumor that is not surgically accessible.	[88]
Adult Low-Grade Gliomas	Small or deep-seated tumors that are not amenable to resection or as an adjunct to surgical resection to improve the extent of resection.	[89,90,91]
High-Grade Gliomas (Grade III/IV)	As a salvage therapy for recurrent tumors or as a palliative therapy for patients with poor performance status.	[92,93,94]

## Data Availability

Not applicable.

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
