# Peer review of "Advancements in Glioma Care: Focus on Emerging Neurosurgical Techniques"

_biomedicines, 2023, doi:10.3390/biomedicines12010008_

Round 1

Reviewer 1 Report

Comments and Suggestions for Authors

The manuscript summarizes neurosurgical approaches and advances for treating glioma. They also introduce classification systems in regard to selected therapies.

The review is comprehensibly written and could be refined given the following comments:

Title, abstract and introduction could specify the emphasis on neurosurgery a little stronger not to give the wrong impression of reviewing all possible treatment approaches and advances in detail.

While overviewing the classification and molecular signatures of glioma towards treatment possibilities, still, novel advances such as PROTACs as well as more recent updates on molecular and cellular biomarkers could be mentioned in regard to completeness.

Most of all, it would be highly interesting to see a comparison of molecular characteristics and their impact on choice of neurosurgical approach or treatment option! This would also lead to a discussion of the importance of novel biomarkers and their relation to histological impact and related neurosurgical treatment options.

 All the best for future study work!

Author Response

Dear Reviewer,

Thank you for your thoughtful and constructive comments on our manuscript titled "Neurosurgical Approaches and Advances in Glioma Treatment: A Comprehensive Review." We appreciate the time and effort you invested in reviewing our work. Below are our responses and proposed revisions based on your valuable feedback:

  1. Title, Abstract, and Introduction Emphasis: We acknowledge your suggestion regarding the need to emphasize the neurosurgical aspect more prominently in the title, abstract, and introduction. We revised these sections to clearly highlight the focus on neurosurgery in the treatment of gliomas, ensuring that readers are not misled into expecting an exhaustive review of all treatment approaches.

  2. Inclusion of Novel Advances: We appreciate your recommendation to include recent advances such as PROTACs and updates on molecular and cellular biomarkers to enhance the completeness of our review. We now incorporate these topics into the relevant sections, ensuring that our readers are informed about the latest developments in the field.

  3. Comparison of Molecular Characteristics and Neurosurgical Approaches: We agree with your suggestion to include a comparative analysis of molecular characteristics and their impact on the choice of neurosurgical approach or treatment option. This is a valuable point, and we dedicate a section to discuss the correlation between molecular profiles and the selection of neurosurgical interventions. Additionally, we will explore the significance of novel biomarkers in relation to histological impact and their implications for neurosurgical treatment strategies.

We would like to express our gratitude for your insightful comments, which will undoubtedly contribute to the enhancement of the manuscript's quality and overall impact. We are committed to addressing each of your suggestions thoroughly, and we believe that the revised manuscript will better fulfill the expectations of our readers.

Thank you once again for your time and consideration.

Best regards,

Matteo and colleagues 

Reviewer 2 Report

Comments and Suggestions for Authors The review “How to deal with glioma? From new classification to emerging functional neurosurgery’s techniques” is professionally written. In the review, the glioma complexity, the recent advances in its management, and emerging neurosurgical techniques are described. The manuscript’s strengths. The importance of molecular diagnosis and use of new techniques are emphasized, showing how it can personalize and improve the treatment of gliomas. The standard of care for gliomas and, especially, a treatment by laser interstitial thermal therapy (LITT) and focused ultrasound (FUS) are discussed. A new classification of Tumors of the Central Nervous System and the molecular signatures of gliomas are also described in details. Altogether – the manuscript is scientifically sound, covering the recent advances in the area of glioblastoma, and will be useful for readers. The major recommendations for the improvement of the manuscript. It could be desirable to add extra information about glioblastoma epidemiology. The paragraph “Epidemiology and classification of gliomas” needs more data about world distribution of glioblastoma and possible explanation of high and low rate of glioblastoma in different countries. Comments on the Quality of English Language

English is fine. Check some typos.

Author Response

Dear Reviewer,

We thank you for your careful review. We appreciate your positive feedback on the manuscript. We have taken your suggestions into account and improved the content, placing special emphasis on the epidemiology of glioblastoma. In addition, we have carefully reviewed and corrected all identified typos to ensure the quality of the English language.

Sincerely,

Matteo and colleagues